# Correlation of Clinicopathological Characteristics of Breast Carcinoma and Depression

**DOI:** 10.3390/healthcare7030107

**Published:** 2019-09-12

**Authors:** Milena B Ilic, Slobodanka Lj Mitrovic, Milena S Vuletic, Uros M Radivojcevic, Vladimir S Janjic, Vesna D Stanković, Radisa H Vojinovic, Dobrivoje S Stojadinovic, Branimir R Radmanovic, Dalibor V Jovanovic

**Affiliations:** 1Department of Pathology, Faculty of Medical Sciences, University of Kragujevac, 34000 Kragujevac, Serbia; lena.ilic@medf.kg.ac.rs (M.B.I.); milena.vuletic@medf.kg.ac.rs (M.S.V.); wesna.stankovic@medf.kg.ac.rs (V.D.S.); dalekg84@medf.kg.ac.rs (D.V.J.); 2Department of Surgery, Faculty of Medical Sciences, University of Kragujevac, 34000 Kragujevac, Serbia; radivojcevicuros@medf.kg.ac.rs; 3Department of Psychiatry, Faculty of Medical Sciences, University of Kragujevac, 34000 Kragujevac, Serbia; vladadok@medf.kg.ac.rs (V.S.J.); biokg2005@medf.kg.ac.rs (B.R.R.); 4Department of Radiology, Faculty of Medical Sciences, University of Kragujevac, 34000 Kragujevac, Serbia; rhvojinovic@medf.kg.ac.rs; 5Department of Anatomy, Faculty of Medical Sciences, University of Kragujevac, 34000 Kragujevac, Serbia; dobrivoje.stojadinovic@medf.kg.ac.rs

**Keywords:** breast cancer, depression, clinicopathological characteristics, molecular subtype, immunohistochemistry

## Abstract

The prevalence of depression among women with breast cancer (BC) is extremely variable in research studies. The aim of this study was to determine the prevalence of depressive disorder in women suffering from BC as well as to examine its relationship with clinical–pathological and immunophenotypic characteristics of BC. The study included 194 patients with BC who were diagnosed with the disease between 2009 and 2015 in the Clinical Center Kragujevac, Serbia. Pathohistological and immunohistochemical analyses was used on the material obtained after the surgical removal of breast tumors, determining all significant clinical and morphological parameters. The level of depression among the examinees confirmed that the differences in the level of depression between the histological grades were statistically significant. According to the univariate binary logistic regression, the depression of a patient correlates with the category of molecular tumor subtype/Luminal A (*p* < 0.0005), PR expression (*p* = 0.050) and lymphatic invasion (*p* = 0.025). Multivariate binary logistic regression showed that the onset of depression associated with the present molecular subtype of the tumor of a worse prognostic character (*p* = 0.019). Depression is a common disorder in women with breast cancer. The level of depression is correlates with some of the clinicоmorphological and immunophenotypic characteristics of BC.

## 1. Introduction

Breast cancer (BC) is the most common malignant tumor in women, with 458,000 deaths and 1.7 million newly diagnosed cases annually worldwide. It is responsible for the cancer-related deaths in 20% of cases [1]. In Serbia, each day there are 13 new women diagnosed with breast cancer, and four of them do not survive this disease. Conventional BC therapy, which includes surgery, radio and chemotherapy, as well as hormone therapy, shows different success rates depending mostly on clinical and pathological characteristics. Thanks to advanced early detection procedures and more effective treatment, the prognosis of the disease has improved and today 89% of women survive five years from the diagnosis. Due to the increase in the survival rate of women affected by BC, the focus of researchers and clinicians has shifted to monitoring of the quality of life following the complex therapeutic protocols conducted in most cases [2].

Communication of the diagnosis and learning that one suffers from this severe illness has a great influence on the psychological and emotional field of a woman, followed by occurrence of negative and unpleasant feelings such as fear, hopelessness, guilt, despair, and a serious sense of abandonment [3]. Depression is the most common psychological disorder happening in patients with BC [4]. The prevalence of depression among women with BC is extremely variable in research studies, ranging from 1 to 56%, and in some cases reaching a value close to 70% [5]. Sometimes, physical distress, the therapeutic treatment of carcinoma, and the dysfunction caused by the disease jointly make the depression more difficult to detect and treat [6].

The uncertainty related to the therapeutic procedures, physical symptoms, fear of death and relapse, change in feminine appearance, physical expression and sexuality, difficulties in carrying out everyday activities, problems in the family, as well as the lack of emotional support from close ones are sources of serious psychological suffering of a patient with BC [7].

The presence of depression in patients with BC contributes to reduced understanding of the problem and acceptance of the condition, as well as to the ability to overcome the difficulties during the therapeutic process. They have a serious obstacle to achieving social interaction, their suitability for therapeutic procedures is significantly reduced, and their overall quality of life is deteriorated. The psychiatric disorders present in these patients have been proven to cause pain, disturbing physical status, satisfaction, emotional and social well-being, as well as general course and outcome of the disease [7].

An extensive study assumed and proved that depression affects, to a small extent, the overall risk of developing cancer [8]. Behavioral factors have a role in the development of cancer and psychiatric disorders can occur in a patient with cancer due to various psychological factors having a neurophysiological and perceptual character. The mechanisms of interaction between depression and cancer are not known, but it is assumed that one of the possible mechanisms involves pathways of endocrine hormones (e.g., sexual hormones), which at some level can affect 5-hydroxytryptamine (5HT) signaling pathway that is important for depression [9]. This assumption makes researchers pay special attention to carcinomas that develop due to hormones, i.e., hormone-dependent carcinomas (e.g., breast cancer, prostate, endometrium, etc.).

There is a need for research to show the prevalence of depressive symptoms among women with breast cancer as well as to show how depression affects the evolution of cancer. The research studies on the problems of pharmacological and psychotherapeutic approaches in the treatment of depressive disorder in women with breast cancer are rare. There is particular interest in researching possible interactions of drugs administered during an oncological treatment and a possible use of antidepressants, with the ultimate outcome on survival. One study demonstrated a pronounced correlation between the efficiency of Tamoxifen used during the oncological treatment and the presence of depressive symptoms in women with BC [10].

Understanding depression in cancer patients can help to integrate strategies into a regular therapeutic procedure that would reduce the psychological suffering of the diseased and undoubtedly improve overall survival. This opinion is supported by the results of studies that explored a number of clinical parameters in cancer patients who showed signs of depression [11]. 

Little is known about the relationship between depression and clinical–pathological and immunophenotypic characteristics of BC, as well as the possible mechanisms that could explain this interaction. In this study, we aimed to reveal the share of patients with depression among patients with BC, as well as to determine possible associations with the clinical–pathological and immunophenotypic characteristics of the tumor.

## 2. Materials and Methods

### 2.1. Study Design

The aim of this study was to determine the prevalence of depressive disorder in women suffering from BC as well as to examine its relationship with clinical–pathological and immunophenotypic characteristics of BC. The research was approved by the Ethics Committee of the Clinical Center, Kragujevac. This research was conducted as a clinical–experimental, retrospective–prospective study. The study included 194 patients with BC who were diagnosed with the disease between 2009 and 2015 in the Clinical Center in Kragujevac. The women involved in the study were from the region of Western Serbia and Sumadija. Inclusion criteria were: women over 18; women who had signed an informed consent to participate in the study; women who had been diagnosed with BC at least three weeks before conducting a self-assessment survey on depression; and the absence of other malignancy or an acute condition that would affect psychological well-being at the time of the interview. Exclusion criteria were: women with a history of psychiatric illness prior to BC diagnosis; and women with disseminated metastases in the brain at the time of the survey.

### 2.2. Pathohistological Analysis and Immunohistochemical Method

Standard pathohistological analysis and immunohistochemical method were used on the material obtained after the surgical removal of breast tumors (tumorectomy/mastectomy), determining all significant clinical and morphological parameters: tumor size, nodal status, histological type, mitotic index, necrosis, lymphovascular invasion, etc. After that, the immunophenotype was defined according to which BC was classified into one of the following groups: Luminal A (Lum A−, ER+, PR+, HER2–), Luminal B (Lum B–, ER+, PR+, HER2+, any level of Ki67 or ER+, PR+, HER2–, Ki67 > 14%), HER2+ (ER–, PR–, HER2+), triple negative breast cancer (TNBC – ER–, PR–, HER2–). The immunophenotype was defined by using an immunohistochemical method involving a range of antibodies that included ER, PR, HER2, and Ki67 in all cases, as well as, if needed, a variety of different cytokeratins following the recommended protocols, using formalin-fixed, paraffin-embedded tumor samples. According to macro and micromorphological characteristics, the pathological stage of the disease was defined according to the WHO recommendations [12]. The evaluation of the immunohistochemical analysis was done by using a semi-quantitative evaluation of expression, according to well-known standards [13].

### 2.3. Depression Evaluation

The respondents were given one survey, complemented with brief instructions and information about the study. The first group of patients completed surveys before the surgery (preoperatively), the second group when they came for the pathohistological finding (7–20 days after the surgery), and the third group completed the survey after the oncological therapy (regular control). A self-assessment of depression was used (Zung scale). The questionnaires consisted of 20 questions with the option to answer with marks from 1 to 4 (1, A Little of The Time; 2, Some Of The Time; 3, Good Part Of The Time; 4, Most Of The Time), with certain questions (2, 5, 6, 11, 12, 14, 16, 17, 18, and 20) in the questionnaire for depression being inversely scored (1=4, 2=3, 3=2, 4=1). The overall score implied the existence/absence of disorders in the following form: up to 40, normal range; 40–50, mild form; 50–60, moderate form; 60–70, markedform; and 70–80, severe form of depressive disorder requiring hospitalization.

### 2.4. Statistical Analysis of Data

Statistical analysis was performed using SPSS Version 19. Continuous variables were presented as mean ± SD or median (25th percentile–75th percentile). Comparisons between two groups were analyzed by *t* test or Mann-Whitney test. Comparisons between more than two groups were done using ANOVA or Kruskal-Wallis test. Spearman’s coefficient was used to test correlations. Qualitative data were analyzed with χ^2^ test. A univariate and multivariate analysis using logistic regression techniques, including odds ratio (OR), was performed to determine the effects of each factor on the dependent variable (depression). Differences were considered significant at *p* < 0.05.

## 3. Results

### General Characteristics

Of the 194 patients who were involved in the study, the average age was 60.48 ± 12.97 (range from 21 to 85). The average time elapsed since the woman had found out that she was suffering from breast cancer until the survey was 9.67 ± 14.08 months. Thirty-three respondents (17%) were from rural areas, while 161 (83%) lived in the city. More than half of the patients completed secondary school, 64 (33%) were employed, while 38 (19.6%) were unemployed, and 92 (47.4%) were retired. Most of them did not provide information on positive family history of significant psychiatric disorders, and half of them also had chronic illnesses.

Fifty-seven women involved in the study belonged to stage I (32.9%), 84 (48.6%) to stage II, 13 (7.5%) to stage III, and 19 women (11%) to stage IV. The most common histological type of BC, according to histopathological analysis, was invasive ductal carcinoma in 157 patients (84.9%); 21 (11.4%) subjects had a diagnosis of invasive lobular carcinoma, while 7 (3.8%) were diagnosed with other histological types (Table 1).

Histopathological analysis showed that 16.2% of tumors were histological grade 1, 50% were histological grade 2, and 33.8% were histological grade 3. Molecular subclassification of BC identified four subtypes: Lum A, Lum B, HER2 positive tumors, and TNBC with the distribution of the subtypes in our study of 19.6%, 56.2%, 18.3%, and 5.9%, respectively (Table 1). 

Out of the 194 patients, 131 (67.5%) showed signs of a depressive disorder, and 63 (32.5%) did not show such signs. Among the women with signs of the disorder, 53 (27.3%) had signs of mild, 47 (24.2%) moderate, 27 (13.9%) marked form, and 4 (2.1%) severe form of disorders (Table 1).

Statistical data processing showed that sociodemographic characteristics (age of the patient, city–village, employed–unemployed) were not related to the level of depression. It turned out that the time that elapsed since the diagnosis and the level of depression were in negative correlation (r = −0.151, *p* = 0.035).

The analysis of the clinical and pathological characteristics of the tumor and the level of depression among the examinees confirmed that the differences in the level of depression between the histological grades were statistically significant (*p* = 0.040). The differences in the level of depression were also significant between nuclear grades (*p* = 0.011). It was also demonstrated that the patients were more depressed when there were signs of lymphatic invasion in the tumors (*p* = 0.014, Table 2).

Statistical analysis of the data obtained through our research showed that there were differences in the level of depression depending on the whether a patient belonged to certain categories. Namely, the difference in the mean values of the level of depression between histological grade 1 and grade 3 was statistically significant (*p* = 0.046, Figure 1). The differences in the level of depression were also significant between nuclear grades 1 and 3 (*p* = 0.015, Figure 2).

In addition to the correlation between the level of depression and the histological and nuclear grades, a positive correlation between the levels of depression and positive nodal status was observed (r = 0.147, *p* = 0.041) as well.

The difference in depression levels was shown to be significant among all BC immunophenotypes (*p* < 0.0005), except between Luminal B and HER2 positive subtypes. The mean depression level of a patient with a tumor belonging to the Luminal A subtype was 39.80 ± 8.47, for Luminal B 48.70 ± 11.88, for HER2 positive 49.68 ± 13.64, and for TNBC 65.33 ± 6.60 (Figure 3). 

Furthermore, the differences in the level of depression and the proliferative index measured through the Ki67 expression were statistically significant (*p* < 0.0005). The patients whose tumors showed a high immunohistochemical index of proliferation were significantly more depressed than those whose tumors showed lower Ki67 expression (Figure 4).

Our study shows that levels of depression and receptor expression for ER were in negative correlation (r = −0.269, *p* < 0.0005). Specifically, the higher the level of depression, the lower the estrogen (Figure 5).

The patients were divided into two groups according to the level of depression: those with a depression level greater than or equal to 40 (depressed) and those with a depression level of less than 40 (non-depressed). The levels of receptor expression for ER in tumors differed significantly between depressed and non-depressed patients (*p* = 0.023, Figure 6)

Depression was also associated with tumors showing signs of lymphatic invasion (r = 0.014). The patients whose tumors had lymphatic invasion exhibited depression in 73.1% of cases, while the percentage in tumors without lymphatic invasion was 53.5%. Depression was also associated with a molecular tumor subtype (*p* < 0.0005). The smallest percentage of depressed patients (36.7%) belonged to the category of Luminal A subtype, while 100% of the TNBC tumor patients were depressed. Depression also correlates with the the expression of the Ki67 proliferation index (*p* < 0.0005). The smallest percentage of depressed patients was in the category of tumors with low Ki67 (Table 3).

According to the univariate binary logistic regression, the depression of a patient depended on the category of molecular tumor subtype/Luminal A (*p* < 0.0005), PR expression (*p* = 0.050), and lymphatic invasion (*p* = 0.025). Multivariate binary logistic regression showed that the onset of depression depended on the present molecular subtype of the tumor of a worse prognostic character (Luminal B, HER2+, TNBC), (*p* = 0.019, Table 4). See other results of binary logistic regression in the same table.

## 4. Discussion

A number of studies have been conducted on the topic of cancer and depression, and the data obtained show very contradictory findings related to the relationship between these two categories. The first major research studies on this topic began in 1980, followed by a single meta-analysis conducted in 1994, which showed that the odds ratio between depression and subsequent cancer development was 1.14, thus showing that this relationship was small and statistically insignificant. 

Depending on the scales for measuring depression used in studies, the percentage of patients with depression is extremely variable, from an extremely low value of 1%, to over 50% in some studies [5]. The reason for such a significant difference in the results obtained can mostly be explained by the choice of the research type and its design, and partially by the choice of the way in which depression levels were measured. Nevertheless, it has been proven that depressive disorder is 2–3 times more common in patients with cancer than in the general population [14]. Frequent comorbidities of cancer and depression have led to numerous studies examining the molecular basis of these disorders, with a view to more clearly understanding the mechanisms of these disorders and to discovering potential sites for therapeutic action. These studies revealed a number of substances that have been proven to influence tumor cell behavior and the psychological status of patients. One of the tested substances is substance P (SP), present in both nerve and many other tissues. It exerts its effect through the NK receptor(NKR), which is expressed both in the peri and intratumoral tissue and has been shown to influence the modulation of tumor cells (proliferation, progression, angiogenesis, metastasis, etc.) [15]. SP and NKR are known to be widespread in the central nervous system, predominantly in structures of the limbic system, such as the thalamus and amygdala, and are thought to have an important influence in integrating the emotional response to stress, so that the development of depression itself may be associated with alterations to the SP/NKR system, as the level of SP in depressive disorders is significantly elevated [16]. It has also been suggested that proinflammatory cytokines released in the immune response and inflammation may contribute to the high rate of depression in cancer patients. Li and co-authors examined the association of a number of proinflammatory and anti-inflammatory cytokines with depression in breast cancer patients [17]. Most of these studies were not designed to show the direction of cause-and-effect relationships between depression and cancer. 

Dealing with the diagnosis of a disease that seriously threatens to endanger one’s life, such as cancer, is almost always followed by a marked stress response, with the appearance of negative and unpleasant feelings such as fear, hopelessness, guilt, despair, and a serious sense of abandonment [3]. Furthermore, the diagnosis and treatment of cancer are associated with numerous acute and chronic stressors that can and certainly do affect the quality of life. It is therefore not surprising that depression is one of the most common disorders diagnosed in people with cancer, with almost 1/3 of patients experiencing depression symptoms at the moment of diagnosis, while 1/4 have symptoms sufficient to meet the criteria for setting up a clinical diagnosis severe depression [18].

The uncertainty related to the therapeutic procedures, physical symptoms, fear of death and relapse, change in feminine appearance, physical expression and sexuality, difficulties in carrying out everyday activities, problems in the family, as well as the lack of emotional support from close ones are sources of serious psychological suffering of a patient with BC [7]. Women find the breast to be very important: it is a symbol of femininity and sexuality, so surgical removal due to cancer is associated with a sense of loss of sexuality, attraction, and maternity, as well as a negative impact on the overall physical appearance [19]. One study showed that depression was associated with a higher level of anxiety, lower self-esteem, and a distorted image of one’s body expression in women with BC [2].

Prolonged and frequent therapeutic treatments and side effects of radio and chemotherapy can trigger depression in women with BC [20]. The level of anxiety is very high in younger patients and middle-aged patients receiving chemotherapy. There are simultaneous occurrences of several different disorders, such as depression, anxiety (24%), panic (18%), and post-traumatic stress disorder (30%) [21]. The patients who have been diagnosed with these disorders have a longer stay in the hospital and a decrease in the quality of life after leaving the hospital [22].

A systematic literature review of the studies conducted in Iran from 2001 to 2016 showed that in all women, measured by different scales, there was at least a mild form of depression. One study showed that as many as 69.4% of women with BC displayed a form of severe depression [23]. The presence of depression in women with BC varied across different climates, cultures, and geographic locations, so in Asian countries the prevalence was 26% [24], in India 21.5% [25], while in Turkey 27.7% had symptoms of moderate and 19.5% of serious depression [26]. In studies conducted in the United States during 2000, 2005, 2010, and 2012, the prevalence of depression was 26% [27], 10% [28], 42% [29], and 56%, respectively [30]. In European countries, such as Germany, it was shown that 11% of affected women suffered from moderate depression, while 12% showed signs of severe depression [31]. Studies in Greece showed that 54.4% of women with BC experienced some form of depressive disorder [32]. In Italy, a study reported that 18% of patients showed symptoms of depression [33]. Possible causes of this kind of variance in data from these studies in different countries include differences in demographic characteristics, such as the age of subjects, their marital status, time of diagnosis of BC, type and number of therapeutic procedures [34], as well as geographical and cultural differences that certainly affect subjects’ mental and psychosocial states. In our research, out of the 194 patients, 131 (67.5%) showed signs of a depressive disorder, and 63 (32.5%) did not show such signs. Among the women with signs of the disorder, 53 (27.3%) had signs of mild, 47 (24.2%) moderate, 27 (13.9%) marked forms, and 4 (2.1%) severe forms of disorders (Figure 2).

Previous studies, which dealt with the correlation between socio-demographic and clinical–pathological characteristics and depression, showed that the results were highly contradictory. They showed that depression was associated with age, education, and material and marital status, [31] while other studies stated that this association did not exist [35]. The result of our research showed that the association of socio-demographic characteristics (age, employment, place of residence, etc.) and depression in women with BC was not statistically significant.

The presence and level of depression in diagnosed women is greatly influenced by the time that elapses from finding out the diagnosis of BC. Previous studies showed that the level of depression in diseased women was the highest within the first year of diagnosis. Within a period of 5 years, it was considered that depression was significantly influenced by individual factors (e.g., age, previous psychological status, lack of support from the environment) to a significantly larger extent than the disease itself or the factors associated with therapeutic procedures (e.g., the number of lymph nodes affected by metastases, tumor size, the result of biopsy, the treatment of the axillary metastasis complications) [36]. Other studies that observed patients and measured the level of disorder six times in 5 years from the moment of diagnosis showed that the depression rate was the highest in the first year (48%) [37].Vahdadina et al. found that despite the fact that the level of anxiety and depression decreased over time, their values were 38.4% and 32.2%, within 18 months, measured using the Hospital Anxiety and Depression Scale (HADS) [36]. Hopwood et al. performed measurements using the HADS scale in 211 patients with advanced carcinoma and confirmed the presence of anxiety and/or depression in 27% of them [38]. They also confirmed that 155 patients continued to show symptoms and signs of these disorders even after 1–3 months. Based on data from a study in which the level of disorder was measured before diagnosing carcinoma, then postoperatively after 1, 3, 6, and 12 months, it was demonstrated that the depression rate decreased from 40.9% to 27.8% after a year, while the symptoms could return as time passed [39].The data in our study also showed that the level of depression decreased as time elapsed from the moment of diagnosis (r = −0.151, *p* = 0.035).

According to available information, this is one of a few studies carried out on the territory of the Republic of Serbia that investigated the relationship between clinical and pathological characteristics of BC and the presence of depressive disorders in the patients. Statistical analysis of the data obtained through our study showed that differences in depression among diagnosed women exist between the following categories: histological tumor grade, especially grade 1 and 3 (*p* = 0.046); nuclear tumor grade (grade 1 and 3, *p* = 0.015); patients with BC who showed signs of lympho–vascular invasion were significantly more depressed than patients with tumors without these signs (*p* = 0.014); and patients with BC metastases in regional lymph nodes (axillary) showed a higher level of depression (r = 0.147, *p* = 0.041).

Hui et al. showed that higher-grade tumors according to TNM classification (a system to describe the amount and spread of cancer in a patient’s body), as well as metastases in regional lymph nodes, were positively correlated with depression in patients with BC [40]. They also showed that the status of hormonal tumor cell receptors was related to depression, i.e., that the higher degree of expression of ER and PR corresponded to a lower degree of depression, while the proliferative index Ki67 was not correlated with depression [41]. Positive correlation of the proliferation index with the degree of depressive disorder found in our study could be a confirmation of the influence of the tumor immunophenotype group. The study by Guo et al. showed that the presence of metastases in regional lymph nodes was in a positive correlation with depression, and that depression is dependent on the status of hormonal receptors, i.e., that tumors with higher ER and PR expression levels showed higher risk of developing depression [42]. This is opposite to our results that show that the hormone status is negatively, and the proliferation index is positively correlated with the degree of depression.

Our study showed that, regarding the expression of hormonal receptors, the level of depression among the respondents differed between tumors belonging to different molecular subtypes. The differences in depression were observed between all subgroups, except between the subtype Luminal B and HER2 positive, with the lowest degree of depression being exhibited by the patients with Luminal A subtype tumor and the highest in TNBC. The results obtained are just another confirmation of the existence of the relationship between depression and the clinical and pathological characteristics of breast cancer, but molecular mechanisms, as well as the cause-and-effect direction of this relationship remain unknown.

In addition to a large number of studies that have dealt with the problem of depression in women with BC, and a large number of contrary results, it remains unclear whether depression has a direct effect on survival or whether the fact that depressed people reject proposed therapeutic procedures is a consequence [43]. In some studies, which investigated the relationship between depression and gastric cancer, it was found that reactive oxygen radicals could be a potential link [42]. Furthermore, extensive research by Munjoz et al. provide a detailed presentation of the molecular aspect of the substance P/receptor for Neurokinin complex (SP/NKR) on the progression of breast cancer and the possibility of use of drugs that block receptors with expecting positive effects on the prognosis of the disease. At the same time, this complex SP/NKR is also present in depressive disorders, and therefore the relationship and interaction of depression and carcinoma could be explained in this way [44].

## 5. Conclusions

Depression is a common disorder in women with breast cancer. The level of depression is correlates with some of the clinicоmorphological and immunophenotypic characteristics of BC. The exact mechanism of interaction between depression and carcinoma remain to be explained as the topic of future studies, in order to open the way for new cancer treatment strategies to better control the disease and improve overall survival.

## Figures and Tables

**Figure 1 healthcare-07-00107-f001:**
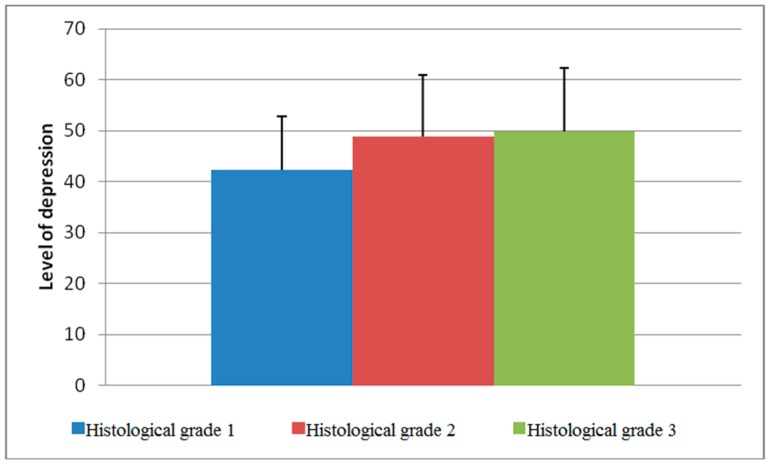
The difference between the mean values of the level of depression between the histological grades.

**Figure 2 healthcare-07-00107-f002:**
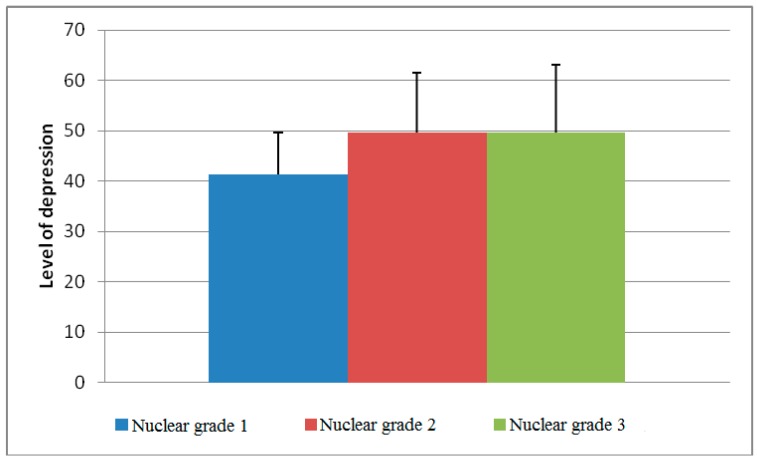
The difference between the mean values of the level of depression between the nuclear grades.

**Figure 3 healthcare-07-00107-f003:**
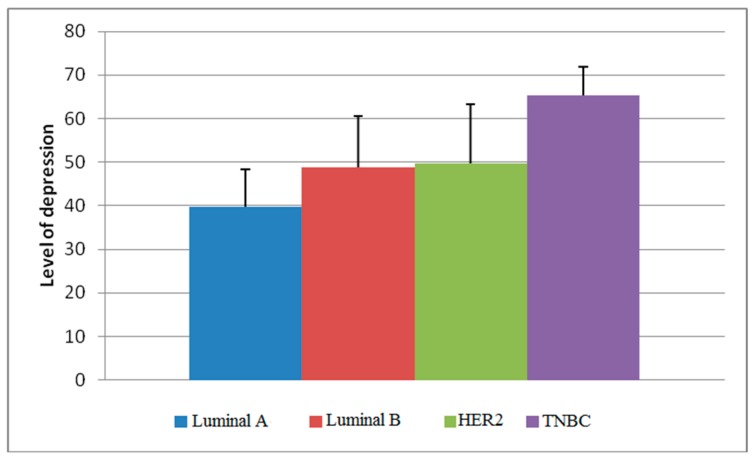
The difference between the mean values of the level of depression between the immunophenotypes.

**Figure 4 healthcare-07-00107-f004:**
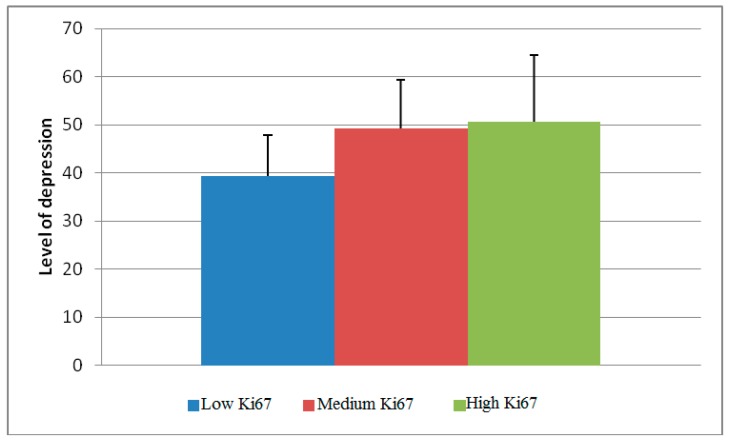
The difference between the mean values of the level of depression between the Ki67 categories.

**Figure 5 healthcare-07-00107-f005:**
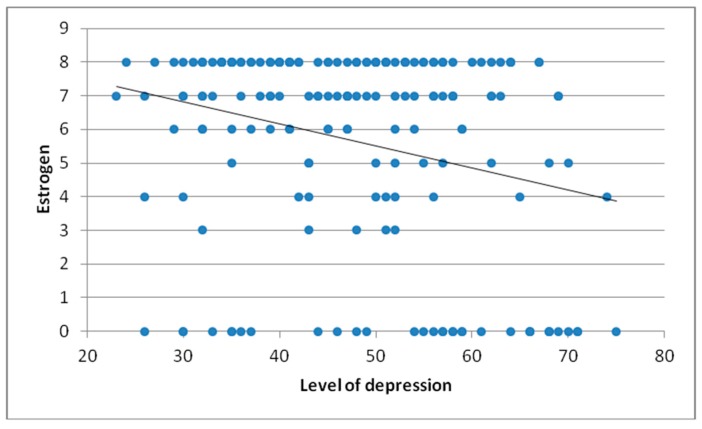
Correlation between level of depression and estrogen.

**Figure 6 healthcare-07-00107-f006:**
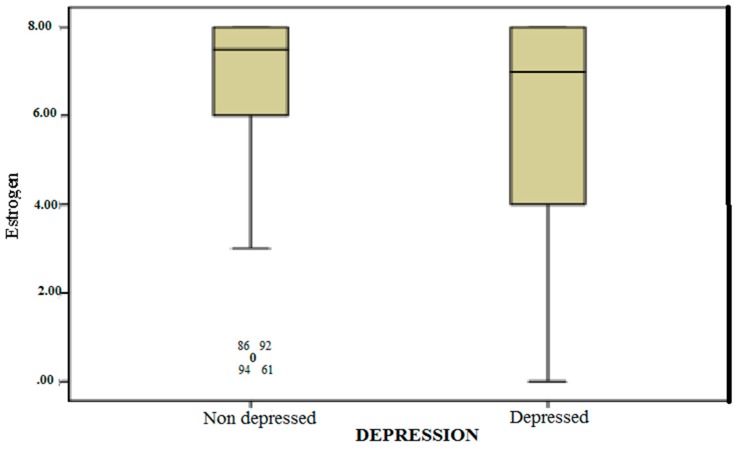
Correlation between level of depression and estrogen.

**Table 1 healthcare-07-00107-t001:** General characteristics of female subjects.

Variable	Mean ± SD or Frequency (%)
Age	60.48 ± 12.97
The time from finding out the diagnosis	9.67 ± 14.08
Urbanity	
village	33 (17%)
town	161 (83%)
Employment	
employed	64 (33.0%)
retired	92 (47.4%)
unemployed	38 (19.6%)
Stage	
1	57 (32.9)
2	84 (48.6%)
3	13 (7.5)
4	19 (11.0)
Histological type	
ductal	157 (84.9%)
lobular	21 (11.4%)
the others	7 (3.8%)
Molecular subtype of breast cancer (BC)	
Luminal A	30 (19.6%)
Luminal B	86 (56.2%)
HER2+	28 (18.3%)
TNBC	9 (5.9%)
Categories of depression	
without depression	63 (32.5%)
mild	53 (27.3%)
moderate	47 (24.2%)
marked	27 (13.9%)
severe	4 (2.1%)

**Table 2 healthcare-07-00107-t002:** Depression level according to clinical and pathological characteristics.

Variable	Level of Depression (mean ± SD)	*p*
Histological grade		
1	42.35 ± 10.43	0.040
2	48.90 ± 11.97	
3	49.77 ± 12.55	
Nuclear grade		
1	41.33 ± 8.30	0.011
2	49.55 ± 12.04	
3	49.56 ± 13.56	
Lymphatic invasion		
No	43.65 ± 11.45	0.014
Yes	49.03 ± 11.85	
Immunophenotypes		
Luminal A	39.80 ± 8.47	<0.0005
Luminal B	48.70 ± 11.88	
HER2	49.68 ± 13.64	
TNBC	65.33 ± 6.60	
Ki67		
low	39.32 ± 8.52	<0.0005
medium	49.18 ± 10.14	
high	50.58 ± 13.92	

**Table 3 healthcare-07-00107-t003:** Characteristics associated with depression expressed as medium.

Variable	Non-Depressed	Depressed	*p*
Estrogen	7.50 (6.00–8.00) *	7.00 (4.00–8.00) *	0.023
Progesterone	6.00 (1.50–7.50) *	4.00 (0.00–7.00) *	0.047
Lymphatic invasion			
No	20 (46.5%)	23 (53.5%)	0.039
Yes	25 (26.9	68 (73.1%)	
Immunophenotypes			
Luminal A	19 (63.3%)	11 (36.7%)	<0.0005
Luminal B	22 (25.6%)	64 (74.4%)	
HER2	9 (32.1%)	19 (67.9%)	
TNBC	0 (0.0%)	19 (100.0%)	
Ki67			
low	19 (67.9%)	9 (32.1%)	<0.0005
medium	4 (11.8%)	30 (88.2%)	
high	24 (31.2%)	53 (68.8%)	

* Values are given as median (25th percentile–75th percentile).

**Table 4 healthcare-07-00107-t004:** Binary logistic regression.

Variable	Univariate Regression	Multivariate Regression
	Odds Ratio	*p*	Odds Ratio	*p*
Age	1.026 (1.103–1.051)	0.029		
Desmoplasia	2.543 (1.400–4.616)	0.002	2.452 (1.233–4.876)	0.011
Progesterone	0.906 (0.821–1.000)	0.050		
Estrogen	0.893 (0.795–1.002)	0.053		
Non luminal A	5.126 (2.128–11.955)	<0.0005	4.454 (1.505–13.178)	0.007
Lymphatic invasion	2.365 (1.112–5.030)	0.025		
Histological type	0.600 (0.325–1.109)	0.103		
Histological grade	1.498 (0.887–2.527)	0.130		
Nuclear grade	1.351 (0.779–2.344)	0.284		
Ki67	1.784 (1.146–2.777)	0.010		
Perineural invasion	1.292 (0.622–2.684)	0.492		
Vascular invasion	1.561 (0.721–3.378)	0.258		
In situ	0.745 (0.401–1.381)	0.350		
HER2	1.376 (0.664–2.853)	0.391		
T	1.226 (0.850–1.768)	0.276		
N	1.373 (0.916–2.058)	0.124		
M	0.630 (0.137–2.904)	0.553

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
