# Peer review of "Correlation of Clinicopathological Characteristics of Breast Carcinoma and Depression"

_healthcare, 2019, doi:10.3390/healthcare7030107_

Round 1

Reviewer 1 Report

The topic is interesting and the manuscript is well written.

However, some improvements are needed:

The enrolled patients answered the survey at different times implying different emotional and physical state (i.e. after surgery, when waiting for the results of the histological examination). Please spend few words for acknowledging this aspect and explaining why (i.e. it was done in order to reach a higher sample in a limited time). There is a long discussion regarding the link between histopathological and biochemical features of cancer and depressive symptoms, but the discussion lacks insights regarding the biological bases that could explain this link. For example, there is a vast literature regarding inflammation and oxidative stress in depression (and cancer), as well as a dysfunctional prostaglandins pathway. Google scholar, search for: "prostaglandins pathway depression cancer";"inflammation depression cancer";"oxidative stress depression cancer"Please choose the articles you find more interesting and improve the discussion.

Author Response

Point 1: The enrolled patients answered the survey at different times implying different emotional and physical state (i.e. after surgery, when waiting for the results of the histological examination). Please spend few words for acknowledging this aspect and explaining why (i.e. it was done in order to reach a higher sample in a limited time). 

Response 1: Оn page 15 number row 377-400 there is an explanation how the level of depression in diseased women changes over the time that has elapsed since the diagnosis was made. For the same reason, we designed our research to test depression at different times.

Point 2: There is a long discussion regarding the link between histopathological and biochemical features of cancer and depressive symptoms, but the discussion lacks insights regarding the biological bases that could explain this link. For example, there is a vast literature regarding inflammation and oxidative stress in depression (and cancer), as well as a dysfunctional prostaglandins pathway. Google scholar, search for: "prostaglandins pathway depression cancer";"inflammation depression cancer";"oxidative stress depression cancer"Please choose the articles you find more interesting and improve the discussion. 

Response 2: Тhank you for the excellent observation. Frequent comorbidities of cancer and depression are the subject of research that  examining the molecular basis of these disorders, with a view to understanding more clearly the mechanisms of these disorders and to discovering potential sites of therapeutic action. Some of the aspects were further presented in the discussion, but with a more detailed explanation we would extend the discussion and thus leave the topic of the paper (page 13, number row 299-315).

Reviewer 2 Report

This paper is of interest due to the interesting hypothesis that it tests. However, has major issues as discussed here:

The paper has used a cross sectional design with about n of 200, with just a few confounders, and then uses very strong causal language. The term depression is dependent upon pathological grade is a very big claim. You can only say there was an association. 

There are too many tables and graphs. I am fine with the bivariate graphs and tables but the pie charts should be deleted and their results should appear in a descriptive table.

The logistic regression should not be limited to the variables that are significant. Why the 2nd model of the last table only has very few variables that are all significant?

The argument of this paper is worse pathology (cancer stage) is associated with depression. What about thinking it this way? People who are depressed may have  tendency to seek care and then be diagnosed at a later  time. Thus, depression predicts stage. You could test this, and could discuss this.

A large body of literature shows that depression operates as a barrier against seeking care in a timely manner.

Author Response

Point 1: The paper has used a cross sectional design with about n of 200, with just a few confounders, and then uses very strong causal language. The term depression is dependent upon pathological grade is a very big claim. You can only say there was an association.

Response 1: Тhank you for the excellent observation. We agree that the term "dependent" is too strong because of the small sample in the survey and that it should be replaced by the term "correlates", and made changes. (page  17, row number 451)

Point 2: There are too many tables and graphs. I am fine with the bivariate graphs and tables but the pie charts should be deleted and their results should appear in a descriptive table.

Response 2: In accordance with the reviewer's comment, we made changes by moving the data from pie 1 and 2 to table 1. (page 6 and 7, number row 190-204)

Point 3: The logistic regression should not be limited to the variables that are significant. Why the 2nd model of the last table only has very few variables that are all significant?

Response 3: In accordance with your comment, other variables that have been statistically processed have been added to the logistic regression table 4. (page 12, number row )

Point 4: The argument of this paper is worse pathology (cancer stage) is associated with depression. What about thinking it this way? People who are depressed may have  tendency to seek care and then be diagnosed at a later  time. Thus, depression predicts stage. You could test this, and could discuss this.

A large body of literature shows that depression operates as a barrier against seeking care in a timely manner.

Response 4: We agree with your comment that a patient's associated depression can be a barrier to going to the doctor and making a timely diagnosis, and that this may be the cause of the association between depression and worse tumor stage. However, the relationship between depression and many other, independent tumor categories (histological and nuclear grade, lymphatic and vascular invasion, hormone receptor status, Ki 67 proliferative index, etc.) was investigated in the paper. It is important to mention that previously diagnosed and treated depression in the women studied was excluding criterion for the study. The aim of this study is to detect depression in cancer patients, due to the possible interactions of the molecular mechanisms of these two processes and to improve the therapy and prognosis of the disease.

Round 2

Reviewer 1 Report

The authors followed the reviewers' suggestions. I do not have further comments to make.

Author Response

Thank you for your comments.
Best regards

Reviewer 2 Report

Some but not all of the revision is satisfactory.

Two of the comments are not taken seriously enough.

First: I had asked the authors to avoid causal language. They have not. They have just made some changes but much more is needed. Here is one example (of many more):

Women with BC show a higher risk of 40 developing depression. The level of depression is dependent on the 41 clinicоmorphological and immunophenotypic characteristics of BC. 

Second: Controlling for stage does not rule out what causes what. That is basic methodology. You can not argue what impacted what with this data, despite your covariates. So, you can only say these things were correlated. YOUR PAPER SHOULD NOT IMPLY DEPRESSION COMES AFTER SOME TYPE OF CANCER. You simply do nt know.

I will give this paper only one more chance, and I either accept (all issues resolved) or reject (same problems continue).

Good luck with a full revision on these two issues.

Round 3

Reviewer 2 Report

The changes are satisfactory.

Language edit is needed.

For example, in a few occasions, the authors say:

The level of depression is correlates with ...

This is not grammatical.